# The Determinants of the 13-Year Risk of Incident Atrial Fibrillation in a Russian Population Cohort of Middle and Elderly Age

**DOI:** 10.3390/jpm12010122

**Published:** 2022-01-17

**Authors:** Marina Shapkina, Andrey Ryabikov, Ekaterina Mazdorova, Anastasia Titarenko, Ekaterina Avdeeva, Elena Mazurenko, Lilia Shcherbakova, Hynek Pikhart, Martin Bobak, Sofia Malyutina

**Affiliations:** 1Institute of Internal and Preventive Medicine—Branch of Federal State Budgeted Research Institution, “Federal Research Center, Institute of Cytology and Genetics, Siberian Branch of the Russian Academy of Sciences”, 630090 Novosibirsk, Russia; a_ryabikov@hotmail.con (A.R.); mazdorova@mail.ru (E.M.); titav@inbox.ru (A.T.); avdeeva_08@inbox.ru (E.A.); poltorackayaes@gmail.com (E.M.); 9584792@mail.ru (L.S.); smalyutina@hotmail.com (S.M.); 2Federal State Budget Educational Institution of Higher Education, Novosibirsk State Medical University of the Ministry of Health of the Russian Federation, 630091 Novosibirsk, Russia; 3Institute of Epidemiology and Health Care, University College London, London WC1E6BT, UK; h.pikhart@ucl.ac.uk (H.P.); m.bobak@ucl.ac.uk (M.B.)

**Keywords:** HAPIEE study, atrial fibrillation, atherosclerosis, arterial hypertension, obesity, diabetes mellitus, aging, determinants, prevalence, Russian population cohort, Cox regression analysis

## Abstract

Atrial fibrillation (AF) is the most common arrhythmia and a predictor of the complications of atherosclerotic cardiovascular diseases (ASCVDs), particularly thromboembolic events and the progression of heart failure. We analyzed the determinants of the 13-year risk of incident AF in a Russian population cohort of middle and elderly age. A random population sample (*n* = 9360, age 45–69 years) was examined at baseline in 2003–2005 and reexamined in 2006–2008 and 2015–2017 in Novosibirsk (the HAPIEE study). Incident AF was being registered during the average follow-up of 13 years. The final analysis included 3871 participants free from baseline AF and cardiovascular disease (CVD) who participated in all three data collections. In a multivariable-adjusted Cox regression model, the 13-year risk of AF was positively associated with the male sex (hazard ratio (HR) = 2.20; 95% confidence interval (CI) 1.26–3.87); age (HR = 1.10 per year; 95% CI 1.07–1.14); body mass index (BMI), (HR = 1.11 per unit; 95% CI 1.07–1.15); systolic blood pressure (SBP), (HR = 1.02 per 1 mmHg; 95% CI 1.01–1.02), and it was negatively associated with total cholesterol (TC), (HR = 0.79 per 1 mmol/L; 95% CI 0.66–0.94). In women, the risk of AF was more strongly associated with hypertension (HT) and was also negatively related to total cholesterol (TC) level (HR = 0.74 per 1 mmol/L; 95% CI 0.56–0.96). No independent association was found with mean alcohol intake per drinking occasion. These results in a Russian cohort have an implication for the prediction of AF and ASCVD complications in the general population.

## 1. Introduction

With the worldwide increase in life expectancy, aging-related conditions are becoming increasingly important. Atrial fibrillation (AF) is the most common arrhythmia, worldwide burden, and an important predictor of the complications of atherosclerotic cardiovascular diseases (ASCVDs), particularly thromboembolic events and progression of heart failure [1,2,3,4]. The prevalence of AF is increasing, and it is expected to become an epidemic in the coming decades, as the population is aging [2,5,6,7,8,9], and will reach up to 14–17 million AF patients by 2030 in the European Union [10].

AF is most prevalent in multimorbid patients with ASCVDs (such as coronary artery disease and stroke), hypertension, heart failure, valvular heart disease, obesity, diabetes mellitus, or chronic kidney disease [5,11,12,13,14,15,16]. About 20–30% of patients with ischemic stroke have AF diagnosed before, during, or after the initial event [17,18,19]. Cognitive impairment [20,21,22], decreased quality of life [23,24], and depressed mood [25] are common in patients with AF. In addition, AF increased the risk of dementia in those without stroke (RR: 1.67; 95% confidence interval (CI): 1.17, 2.38) [26].

The prognostic importance of AF underlined the development and regular update of guidelines for the management of AF [2,3], but the asymptomatic and paroxysmal course often allows only a retrospective verification of AF as a mechanism of complications that have already arisen. Consequently, preclinical assessment of AF risk and the identification of patterns of AF predictors is a priority.

For this report, we investigated the determinants of the 13-year risk of incident AF in a Russian population cohort of the middle and elderly age.

## 2. Materials and Methods

### Study Population

A random population sample (*n* = 9360, age 45–69 years) was examined at baseline in 2003–2005 in Novosibirsk within the Russian arm of the HAPIEE (Health, Alcohol and Psychosocial Factors in Eastern Europe) study [27]. The average follow-up period was 12.8 years (standard deviation (SD) = 0.78, median = 12.7) in men and 12.9 years (SD = 0.77, median = 12.9) in women (until 31 December 2017). During the follow-up, two reexaminations of the cohort were being conducted in 2005–2008 (2nd, *n* = 6031, age 47–72 years) and 2015–2017 (3rd, *n* = 3898, age 55–84 years).

The analysis included respondents with available rest electrocardiogram (ECG) at the survey: 9255 participants at baseline examination and 3878 on the 3rd reexamination.

The design of this study is a prospective cohort study.

The study was approved by the Ethics Committee (Protocol No. 1 from 14 March 2002 and Protocol No. 12 from 8 December 2020) of the IIPM—Branch of IC&G SB RAS (Institute of Internal and Preventive Medicine – Branch of Federal State Budgeted Research Institution, “Federal Research Center, Institute of Cytology and Genetics, Siberian Branch of the Russan Academy of Sciences”). All study participants signed the patient informed consent form.

The research protocol of examinations throughout all stages (waves) included a standard epidemiological assessment of cardiovascular diseases (CVDs) and their risk factors and health parameters. Details of the study protocol have been described elsewhere [27]. In this analysis, we included the following instrumental and laboratory parameters: body mass index (BMI), blood pressure (BP), heart rate, resting ECG, levels of total cholesterol (TC), triglycerides (TGs), high-density lipoprotein cholesterol (HDLC), gamma-glutamyl transpeptidase (GGTP), and glucose in blood serum were measured enzymatically. Low-density lipoprotein cholesterol (LDLC) was calculated using the Friedewald formula.

To assess fasting plasma glucose (FPG), we converted the fasting serum glucose level using the formula of the European Association for the Study of Diabetes (EASD), 2007 [28]:

FPG (mmol/L) = −0.137 + 1.047 × Glucose of serum (mmol/L).

The resting ECG was recorded in 12 standard leads using electrocardiograph Cardiax (IMED Ltd., Budapest, Hungary), the timeframe for resting ECG was 30 s (including two records for 6 leads each, record at inspiration and record “for rhythm”). ECG changes were assessed manually by Minnesota Code (MC) [29] by two readers (S.M., M.S.). For repeatability of ECG, we conducted a double-blind assessment of ECG in a random sample (100 records), the mean agreement coefficient Kappa was 0.85.

HT was established at levels of systolic (SBP) or diastolic (DBP) blood pressure ≥ 140/90 mm Hg according to European Society of Cardiology (ESC) Guidelines, 2018 [30], and/or under antihypertensive treatment during the last 2 weeks.

Diabetes mellitus (DM) was defined in the presence of a history of diabetes with glucose-lowering treatment and/or by fasting plasma glucose level of ≥7 mmol/L [31].

The presence of coronary artery disease (CHD) was determined according to epidemiological criteria based on (1) survey data (positive response to the Rose questionnaire for angina pectoris and/or “ischemic” ECG changes (1, 4, 5 MC classes) or (2) medical history of myocardial infarction, or acute coronary syndrome, or coronary artery bypass grafting, or percutaneous transluminal coronary angioplasty confirmed by hospitalization.

The presence of stroke was established on the basis of a medical history of stroke or a transient ischemic attack confirmed by hospitalization. The history of chronic heart disease (CHD) and/or stroke was determined as the presence of CVD.

A person who smoked at least one cigarette a day was classified as a smoker. Smoking status was categorized as current smoker, former smoker, and never smoked.

The amount of alcohol consumed was converted to pure ethanol (g). The alcohol consumption in the present study was analyzed by the mean dose per occasion. Marital status was dichotomized as married (or cohabiting) and single (never been married, divorced, or widower/widow). The level of education was dichotomized as high education and education less than high.

The new-onset AF revealed at the reexamination during the follow-up was registered as an endpoint (permanent type) and was defined by Minnesota Code (MC 8-3-1, 8-3-2; 6-8 with “fibrillation” for atria).

Firstly, for the sake of the cross-sectional estimates of the AF prevalence, we included all subjects that have adequate resting ECG at the baseline examination (*n* = 9255) and 3rd examination (*n* = 3878).

Secondly, for cohort analysis of the incident AF, we excluded individuals with prevalent AF (*n* = 146) and CVD at baseline (*n* = 1888), non-responders in any of the follow-up examinations (*n* = 2759), those with technically inadequate ECGs at any wave (*n* = 105), or missing baseline data on the factors analyzed (*n* = 592). Finally, the data on 3871 persons (1775 men and 2096 women) aged 45–69 years at baseline were analyzed for the risk of incident AF. The analysis in the CVD-free population was applied in correspondence with a similar approach in other cohort studies of the AF risk [32,33,34,35].

Thirdly, for the sensitivity analysis, we repeated calculations based on a cohort without exclusion of the baseline CVD.

Statistical analysis was performed using the Statistical Package for Social Sciences (SPSS) version 13.0 for Windows (IBM, Armonk, NY, USA). First, we used ANOVA and nonparametric tests, to compare continuous variables between those who developed AF and their counterparts, and cross-tabulation to compare categorical variables. Second, we used Cox regression (age- and multivariable-adjusted models) to analyze predictors of AF.

Model 1 was age- and sex adjusted in total population and age adjusted in analysis split by sex; model 2 was adjusted for age, sex, BMI, SBP, TC, TG, smoking, alcohol consumption (mean dose per occasion, g) based on the risk factors preselected in model 1; model 3 was adjusted for age, sex, BMI, TC, HT, DM, smoking, alcohol consumption, level of education, and marital status. Hypothesis testing was considered statistically significant at *p*-value < 0.05.

## 3. Results

In cross-sectional analysis, the prevalence of AF in the cohort population increased from 1.6% (*n* = 146/9255) at the age of 45–69 years (1.1% in women and 2.1% in men) to 4.2% (*n* = 164/3878) at the age of 55–84 years (3.0% in women and 6.1% in men) over the 13-year follow-up period (Figure 1).

Among participants free from baseline AF and CVD, 122 new cases of AF were identified, accounting for 3.2% (*n* = 122/3871), 2.6% in women, and 3.8% in men.

The baseline characteristics of participants with incident AF occurred during the follow-up, and those without AF are presented in Table 1. Both men and women with incident AF were older and had stronger expressed cardiometabolic risk factors than their counterparts (higher values of BMI, SBP, and DBP), and higher HT prevalence (among women). There was no difference between groups in blood lipid profile or carbohydrate metabolism parameters. In addition, there was no significant difference in behavioral factors (smoking and alcohol consumption) and in socioeconomic status (educational level and marital status).

In the series of age- and sex-adjusted Cox regression of model 1, 18 risk factors were tested. The 13-year risk of incident AF was positively associated with male sex, BMI, SBP, DBP, or presence of HT, and it was negatively associated with TC and LDLC levels (Appendix A). According to the results of age-adjusted model 1, the following covariates were included for further analysis: age, sex, BMI, SBP, TC, TG, smoking, alcohol consumption (mean dose per occasion, g) in model 2; age, sex, BMI, TC, HT, DM, smoking, alcohol consumption (mean dose per occasion, g), education, and marital status in model 3.

Men in our studied population sample had a 2.2-fold increased risk of AF (95% CI 1.17–3.47), compared with women, regardless of other factors. Thus, further analysis of the contribution of the studied risk factors to the 13-year risk of AF was performed with stratification by sex.

In men (Table 2, Figure 2), in age-adjusted models, the risk of incident AF was positively associated with the value of BMI, SBP, and alcohol intake. After controlling by biological and behavioral factors, the risk of incident AF was positively associated with age (HR = 1.09; 95% CI 1.04–1.13), BMI (HR = 1.10; 95% CI 1.04–1.17), and value of SBP (HR = 1.01; 95% CI 1.001–1.021) in model 2. Some associations attenuated after additional adjustment for HT, DM, education, and marital status in model 3, and AF remained to be associated with age (HR = 1.10; 95%CI 1.06–1.14) and value of BMI (HR=1.11; 95% CI 1.04–1.17), regardless of other factors.

In women (Table 3, Figure 3), in age-adjusted models, the risk of incident AF was positively associated with the value of BMI (HR = 1.10; 95% CI 1.05–1.15), SBP (HR = 1.02; 95% CI 1.01–1.03), or the presence of HT (HR = 2.79; 95% CI 1.30–5.99), and negatively related to TC (HR = 0.70; 95% CI 0.55–0.89) and LDLC (HR = 0.68; 95% CI 0.52–0.88). These relationships remained significant in models 2 and 3 independent of biological, behavioral, and social factors, except for the value of LDLC.

In sensitivity analysis among all subjects, regardless of the prevalent CVD, the presence of baseline CVD was associated with AF risk (HR = 2.25 per year; 95% CI 1.59–3.17). The HRs for associations between primarily revealed determinants and AF slightly attenuated but remained significant; we did not find additional associations between other studied risk factors and AF.

## 4. Discussion

In this Russian population-based cohort, the baseline prevalence of AF was 1.6% at the age of 45–69 years and about 4% at the age of 55–84 years. The frequency of AF increased from 0.3% in the younger age group (45–50 years) up to 13% in men and 6% in women in the older age group (80–84 years). Our results were close to the findings from the North American and European population studies. In the prospective analyses, among the persons free from baseline CVD, the 13-year risk of incident AF was positively associated with male sex, SBP, or the presence of hypertension, and BMI value independent of other factors. In the total cohort, the risk of incident AF was also associated with the prevalent CVD.

AF is an age- and sex-specific condition. In the AnTicoagulation and Risk Factors In Atrial Fibrillation (ATRIA) study (USA), the prevalence of AF was 0.1% among adults younger than 55 years and 9.0% for those aged 80 years and older [5]. The overall prevalence of AF in the Rotterdam study was 5.5%, rising from 0.7% in the age group of 55–59 years to 17.8% in those aged 85 years and above [36].

The profile of AF determinants and the strength of their effects were similar to other studies worldwide. For example, the 38-year follow-up data from the Framingham study (FHS) showed that men were 1.5-fold more likely to develop AF than women, adjusted for age and predisposing conditions [13]. In our study, over a 13-year follow-up period, being male had an approximately twofold increased risk of incident AF, independent of other risk factors.

The FHS data also demonstrated that age is the strongest risk factor for AF when compared with other factors, including male sex, BMI, diabetes, smoking, alcohol intake, SBP, heart failure, and myocardial infarction [37]. In our study, age also strongly contributed to the risk of incident AF and increased the risk of AF by 10% each year.

With regard to elevated BP, in the studied Russian population sample, those with incident AF during follow-up had higher baseline SBP and DBP, compared with those free from this condition, and this relationship was stronger among women. In women, the presence of HT was a powerful predictor of AF and independently associated with the risk of AF, increasing by 2.3 times, compared with normotensives. The value of SBP was positively associated with the risk of AF in both sexes. The Cohorts for Heart and Aging Research in Genomic Epidemiology model for atrial fibrillation (CHARGE-AF) consortium also found that both systolic and diastolic BP were predictors of AF risk [38]. In the FHS, the adjusted risk for AF was slightly weaker and amounted to 1.4- and 1.5-fold risk for women and men with HT, respectively [13].

Long-term (13-year) trajectories of systolic blood pressure and hypertension in the Tromsø study were also associated with an increased incidence of AF. However, this association was stronger in women with baseline elevated systolic blood pressure and doubled the risk of AF incidence, regardless of the dynamics of SBP during the follow-up period [39].

Elevated BMI is both an independent risk factor for AF and a predisposing for the development of HT, DM, and CVD. In our population cohort, those who developed AF during the 13-year follow-up period had higher baseline BMI values (*p* < 0.001); an increase in BMI by 1 kg/m^2^ increased the risk of AF by 10% in both, men and women, independent of other factors. The association between the value of BMI and increased risk of developing AF was reported in numerous population studies [40,41,42]. For example, in the Danish Diet, Cancer, and Health Study, the adjusted hazard ratio for AF per unit of increase in the body mass index was 1.08 (95% CI: 1.05–1.11) in men and 1.06 (95% CI: 1.03–1.09) in women [36]. A dose–response relationship was observed, with each one-unit increase in BMI associated with a 3–4.7% increase in AF risk [40,43,44].

The blood lipid fractions have a complex relationship with AF. In our analysis, LDLC and TC values were negatively related to the 13-year risk of incident AF in age-adjusted models; for TC, this association remained significant after controlling for other risk factors and was mainly confer to women. In a combined analysis of data from the Multi-Ethnic Study of Atherosclerosis (MESA) and the FHS, on the contrary, a high level of HDLC was inversely associated with AF risk (HR 0.64, 95% CI 0.48–0.87 in those with levels ≥60 mg/dL versus <40 mg/dL), whereas high TG value was associated with a higher risk of AF (HR = 1.60, 95% CI 1.25–2.05 in those with levels ≥200 mg/dL versus <150 mg/dL). TC and LDLC were not related to the risk of AF in the referred analysis [45]. In contrast, the data on the Atherosclerosis Risk in Communities (ARIC) study showed that the high levels of LDLC and TC were associated with a lower risk of AF, whereas HDLC and TG were not related to AF risk [46]. A similar inverse association between LDLC and the risk of AF was found in the Women’s Health Study [47]. The mechanisms of this inverse association between atherogenic lipids and AF are not fully understood and might be related to the complex relationship between the cholesterol and the stabilizing effect on myocardial membranes and ion channel function [48,49,50], along with its role in chronic inflammation and oxidative stress [51]. In addition, AF is an age-associated condition, and blood lipid levels generally decrease in patients of the age of over 60. Additionally, subclinical hyperthyroidism with reduced lipid levels may be an independent risk factor for AF [52].

In general, the identified cluster of AF predictors, such as obesity, hypertension, age, and male sex, make it possible to consider AF as an atherosclerotic cardiovascular condition. Some differences in AF determinants or coefficients between studies might be due to various study designs and approaches in the diagnosis of AF, as well as racial characteristics, morbidity, and risk factors profile, socio-economic characteristics, medical care, and lifestyle in the studied populations.

### Study Limitations

The present study has some limitations. The incident AF cases in the analysis were limited to permanent AF form recorded by resting ECG in repeated examination and did not include the cases of paroxysmal AF that occurred during the follow-up, including the fatal ones. This might lead to underestimation of risk coefficients but is unlikely to change the identified determinants of AF.

We did not find associations between incident AF and behavioral factors, which can also be related to the fact that we included only cases of permanent AF.

To overcome this shortage, at the next stage of analysis, we plan to update the completeness of AF endpoints by ascertainment of paroxysmal and persistent AF cases, both fatal and non-fatal, during the 13-year follow-up from CVD and mortality registers. Upon data completeness, we plan further post hoc analyses.

Another limitation might be related to the approach that only respondents free from baseline AF and CVD were included in the analysis of AF determinants. This is a common practice in the cohort studies for cardiovascular outcomes including the risk of AF (for example, Framingham Heart study [32], Renfrew/Paisley study [33], nation-wide cohort study of Swedish [34], REasons for Geographic And Racial Differences in Stroke (REGARDS) study [35]), and the using of this approach allows us to compare results. Moreover, to assess the overall burden of AF, we additionally performed a sensitivity analysis of the determinants of the risk of incident AF among all respondents, regardless of the presence of CVD. As expected, baseline CVD was associated with the AF risk; the HRs for associations between primarily revealed determinants and AF slightly attenuated but remained significant. Additionally, we did not find additional associations between the studied risk factors and AF. Thus, the final results practically did not change (Appendix A).

The last limitation relates to non-response or attrition bias where persons more prone to AF risk due to CVD, arrhythmia, or other health problems did not attend the examination. However, we addressed the cohort free from baseline CVD and AF, and this issue has no impact on the baseline results. Non-response for reexamination limited the number of incident AF cases registered, but it is unlikely that individuals with chronic conditions as permanent AF would be systematically more prone to not attending the examination. Moreover, in separate analyses, we observed that non-responders have better health than responders (personal communication with D. Denissova [53]).

At the same time, according to our knowledge, this is the first cohort study in Russia that explored the risk of AF and identified the predictors and individual risk coefficients for incident AF at the population level.

## 5. Conclusions

In this Russian population-based cohort of middle-aged and older individuals, the prevalence of AF registered by resting ECG increased from 1.6% to 4.2% with the age from 45–69 to 55–84 years. The incidence rate of new-onset AF cases among persons free from baseline CVD was 3.2%. The 13-year risk of incident AF was positively associated with male sex, age, BMI, SBP, or the presence of hypertension, and was negatively associated with total cholesterol level. In women, the 13-year risk of incident AF was more strongly associated with HT and inversely associated with total cholesterol levels.

The present study was limited by the consideration of largely permanent AF based on the resting ECG and did not include the cases of paroxysmal AF during the follow-up. Further studies are needed to cover the overall burden of AF at longitudinal follow-up in a population setting.

At the same time, the findings in this Russian cohort are generally consistent with the data obtained in the European and North American population studies. The findings of individual risk of AF in the Russian cohort have implications for the prediction of AF and ASCVD complications.

## Figures and Tables

**Figure 1 jpm-12-00122-f001:**
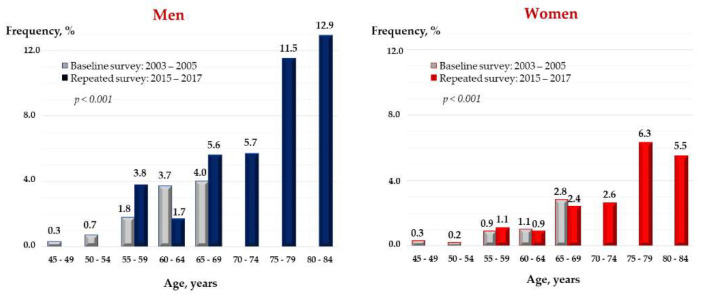
Prevalence of AF 2003–2017 by age groups in 2003–2005 and 2015–2017. The HAPIEE study, Russian population cohort. AF, atrial fibrillation.

**Figure 2 jpm-12-00122-f002:**
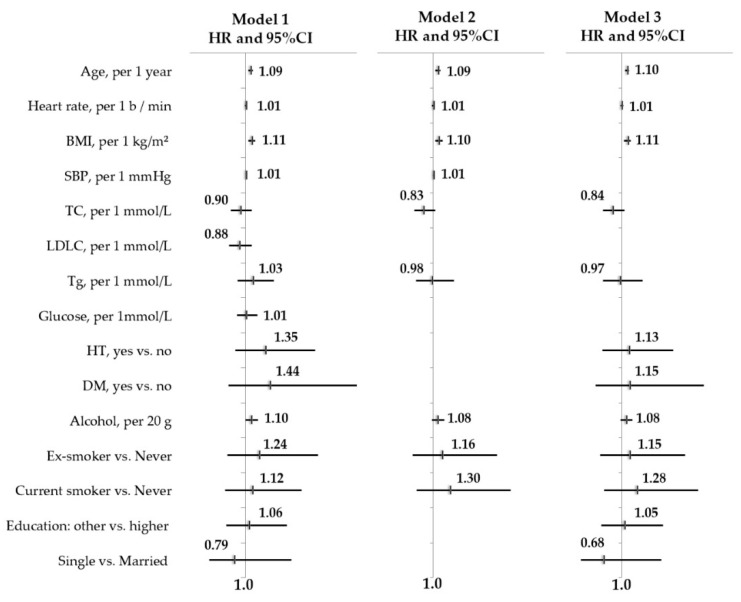
Determinants of 13-year risk of incident AF in men free from baseline AF and CVD. Cox regression analysis, age- and multivariable-adjusted models. The HAPIEE study, Russian population cohort, *n* = 1775. AF, atrial fibrillation; CVD, cardiovascular disease; BMI, body mass index; SBP, systolic blood pressure; TC, total cholesterol; LDLC, low-density lipoprotein cholesterol; TG, triglycerides; HT, hypertension; DM, diabetes mellitus; HR, hazard ratio; CI, confidence interval.

**Figure 3 jpm-12-00122-f003:**
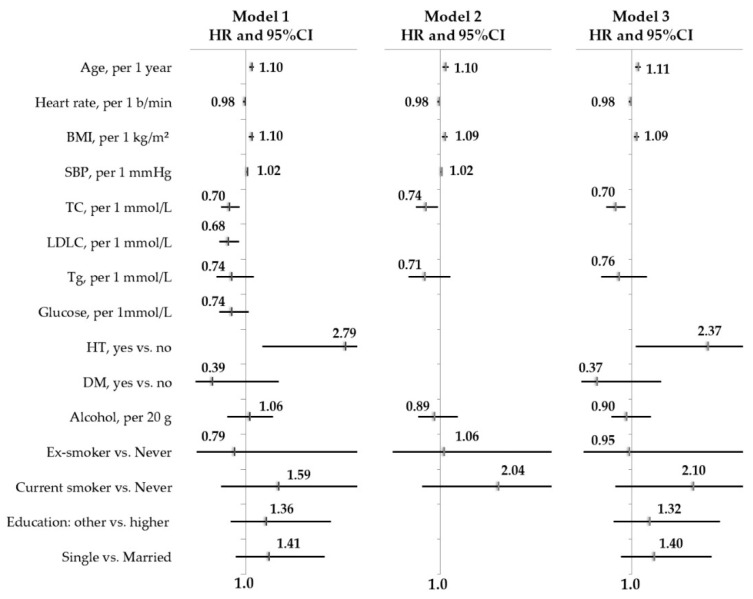
Determinants of 13-year risk of incident AF in women free from baseline AF and CVD. Cox regression analysis, age- and multivariable-adjusted models. The HAPIEE study, Russian population cohort, *n* = 2096.

**Table 1 jpm-12-00122-t001:** Sex-specific characteristics of baseline sample depending on incident AF. The HAPIEE study, Russian population cohort free from baseline AF and CVD, *n* = 3871.

Risk Factors	Men, Mean (SD)/*n* (%)			Women, Mean (SD)/*n* (%)		
	iAF(−) *n* = 1707	iAF(+) *n* = 68	*p* ^1^	iAF(−) *n* = 2042	iAF(+) *n* = 54	*p* ^2^
Age, years	57.6 (6.99)	60.6 (6.30)	<0.001	57.2 (6.95)	61.4 (5.76)	<0.001
Heart rate, b/min	71.1 (12.48)	70.7 (12.16)	0.807	71.2 (10.33)	68.7 (10.7)	0.078
BMI, kg/m^2^	26.0 (4.13)	28.0 (4.25)	<0.001	29.5 (5.42)	32.5 (5.70)	<0.001
SBP, mmHg	142.3 (22.98)	151.0 (24.89)	0.002	141.5 (24.89)	158.9 (27.46)	<0.001
DBP, mmHg	89.8 (13.18)	93.1 (13.29)	0.045	89.0 (13.01)	96.1 (15.40)	<0.001
HT	1002 (58.7%)	45 (66.2%)	0.134	1272 (62.3%)	46 (85.2%)	<0.001
DM	152 (9.0%)	8 (12.1%)	0.253	175 (8.7%)	2 (3.8%)	0.160
TC, mmol/L	5.9 (1.21)	5.9 (1.21)	0.585	6.5 (1.26)	6.2 (1.12)	0.077
LDLC, mmol/L	3.8 (1.08)	3.7 (1.01)	0.690	4.2 (1.13)	3.9 (0.98)	0.060
HDLC, mmol/L	1.5 (0.38)	1.5 (0.30)	0.230	1.6 (0.34)	1.6 (0.37)	0.521
TG, mmol/L	1.4 (0.75)	1.5 (0.69)	0.514	1.5 (0.82)	1.4 (0.76)	0.403
GGTP, U/L	38.3 (45.03)	35.1 (20.01)	0.566	28.1 (30.40)	28.7 (17.15)	0.891
Glucose, mmol/L	5.9 (1.49)	5.9 (0.87)	0.916	5.9 (1.39)	5.6 (0.61)	0.193
Alcohol per occasion, g	55.4 (45.89)	62.5 (48.11)	0.216	22.0 (16.83)	20.4 (12.89)	0.482
Smoking status:						
Never smoked	458 (26.8)	21 (30.9)	0.215	1756 (86.0)	49.0 (90.7)	0.559
Former smoker	390 (22.9)	20 (29.4)		87 (4.3)	1 (1.9)	
Current smoker	858 (50.3)	27 (39.7)		199 (9.7)	4 (7.4)	
The level of education:			0.257			0.212
Higher	575 (33.7)	26 (38.2)		615 (30.1)	13 (24.1)	
Other	1132 (66.3)	42 (61.8)		1427(69.9)	41 (75.9)	
Marital status:						
Married	1495 (87.6)	62 (91.2)	0.250	1255 (61.5)	27.0 (50.0)	0.060
Single	212 (12.4)	6 (8.8)		787 (38.5)	27,0 (50.0)	

^1^*p*-value between groups depending on incident AF in men; ^2^*p*-value between groups depending on incident AF in women; iAF(+) or iAF(−)—a presence or an absence incident AF, respectively. AF, Atrial fibrillation; CVD, cardiovascular disease; SD, standard deviation; iAF, incident AF; BMI, body mass index; SBP, systolic blood pressure; DBP, diastolic blood pressure; HT, hypertension; DM, diabetes mellitus; TC, total cholesterol; LDLC, low-density lipoprotein cholesterol; HDLC, high-density lipoprotein cholesterol; TG, triglycerides; GGTP, gamma-glutamyl transpeptidase.

**Table 2 jpm-12-00122-t002:** Associations between risk factors and 13-risk of incident AF in men free from baseline AF and CVD. Cox regression analysis, age- and multivariable-adjusted models.

Risk Factors	Model 1 HR (95% CI)	Model 2 HR (95% CI)	Model 3 HR (95% CI)
Age, per 1 years	1.09 (1.05–1.13)	1.09 (1.04–1.13)	1.10 (1.06–1.14)
Heart rate, per 1 b/min	1.01 (0.99-1.03)	1.00 (0.98–1.03)	1.01 (0.99–1.03)
BMI, per 1 kg/m^2^	1.11 (1.05–1.17)	1.10 (1.04–1.17)	1.11 (1.04–1.17)
SBP, per 1 mmHg	1.01 (1.00–1.02)	1.01 (1.00–1.02)	
TC, per 1 mmol/L	0.90 (0.73–1.11)	0.83 (0.66–1.04)	0.84 (0.67–1.05)
LDLC, per 1 mmol/L	0.88 (0.70–1.11)		
TG, per 1 mmol/L	1.13 (0.85–1.51)		0.97 (0.67–1.40)
Glucose, per 1 mmol/L	1.01 (0.84–1.21)		
GGTP, per 1 U/L	1.00 (1.00–1.01)		
HT, yes vs. no	1.35 (0.81–2.25)		1.13 (0.66–1.93)
DM, yes vs. no	1.44 (0.69–3.01)		1.15 (0.53–2.48)
Alcohol, per 20 g	1.10 (1.00–1.22)	1.08 (0.98–1.20)	1.00 (1.00–1.01)
Smoking status:			
- Former smoker vs. Never	1.24 (0.67–2.23)	1.16 (0.63–2.15)	1.15 (0.61–2.15)
- Current smoker vs. Never	1.12 (0.63–2.01)	1.30 (0.70–2.40)	1.28 (0.69–2.38)
Level of education:			
- Other vs. higher	1.06 (0.65–1.74)		1.05 (0.63–1.75)
Marital status:			
- Single vs. married	0.79 (0.34–1.82)		0.68 (0.27–1.72)

HR, hazard ratio; CI, confidence interval; model 1 adjusted for age; model 2: adjusted for age, BMI, SBP, TC, TG, smoking, alcohol consumption; model 3 adjusted for age, BMI, TC, HT, DM, smoking, alcohol consumption, education, and marital status.

**Table 3 jpm-12-00122-t003:** Associations between risk factors and 13-risk of incident AF in women free from baseline AF and CVD. Cox regression analysis, age- and multivariable-adjusted models.

Risk Factors	Model 1 HR (95% CI)	Model 2 HR (95% CI)	Model 3 HR (95% CI)
Age, per 1 years	1.10 (1.06–1.15)	1.09 (1.06–1.12)	1.10 (1.07–1.14)
Heart rate, per 1 b/min	0.98 (0.95–1.00)	0.98 (0.95–1.00)	0.98 (0.95–1.00)
BMI, per 1 kg/m^2^	1.10 (1.05–1.15)	1.09 (1.04–1.14)	1.09 (1.04–1.14)
SBP, per 1 mmHg	1.02 (1.01–1.03)	1.02 (1.01–1.03)	
TC, per 1 mmol/L	0.70 (0.55–0.89)	0.74 (0.57–0.96)	0.70 (0.54–0.89)
LDLC, per 1 mmol/L	0.68 (0.52–0.88)		
TG, per 1 mmol/L	0.74 (0.47–1.15)		0.76 (0.45–1.28)
Glucose, per 1 mmol/L	0.74 (0.52–1.05)		
GGT, per 1 U/L	1.01 (1.00–1.01)		
HT, yes vs. no	2.79 (1.30–5.98)		2.37 (1.07–5.22)
DM, yes vs. no	0.39 (0.09–1.60)		0.37 (0.09–1.53)
Alcohol, per 20 g	1.06 (0.67–1.49)	0.89 (0.60–1.32)	0.90 (0.63–1.35)
Smoking status:			
- Former smoker vs. Never	0.79 (0.11–5.79)	1.06 (0.14–7.93)	0.95 (0.13–7.11)
- Current smoker vs. Never	1.59 (0.55–4.64)	2.04 (0.68–6.15)	2.10 (0.70–6.27)
Level of education:			
- Other vs. Higher	1.36 (0.73–2.54)		1.32 (0.67–2.58)
Marital status:			
- Single vs. Married	1.41 (0.82–2.42)		1.40 (0.81–2.44)

HR, hazard ratio; CI, confidence interval; model 1 adjusted for age; model 2 adjusted for age, BMI, SBP, TC, TG, smoking, alcohol consumption; model 3 adjusted for age, BMI, TC, HT, DM, smoking, alcohol consumption, education, and marital status.

## Data Availability

The data presented in this study are available in tabulated form on request. The data are not publicly available due to ethical restrictions and project regulations.

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
