# Peer review of "The Determinants of the 13-Year Risk of Incident Atrial Fibrillation in a Russian Population Cohort of Middle and Elderly Age"

_jpm, 2022, doi:10.3390/jpm12010122_

Round 1

Reviewer 1 Report

In the present manuscript, Shapkina and co-workers aimed to prospectively  evaluate the  13-year AF risk determinants in a random  Russian population cohort, aged between 45-69 years,   being the Russian arm of the HAPIEE project.  The authors should be acknowledged for the effort put into the manuscript. Despite the fact that AF detection was limited to a single 12-lead ECG registration, that almost certainly decreased the AF incidence rate,  I find interesting and novel. However, there are several caveats that should be corrected.

  1. The major issue: The authors have excluded from the analysis individuals with cardiovascular disease (CVD) (n=1888/9360)  detected at the initial medical check-up. Therefore they actually assessed  the 13-year AF risk determinants not in a random  Russian population cohort but in a cohort free from CVD. As far as I am concerned this issue had a significant impact on the presented results and conclusions, especially that the authors claim further in the manuscript that “the results in the presented cohort have the implication for prediction of AF in a general population”. I cannot find a rationale for this exclusion criterion. It is a well known fact that  AF prevalence is strongly associated with concomitant CVD so the overall AF burden is probably underestimated.  Moreover the authors provide the number of initially AF detected cases (1.6%, 145/9360) in the full study cohort, including individuals with CVD. Therefore I have doubts whether “the prevalence of AF in the study  cohort increased from 1.6% to 4.2%”  if the new AF cases were detected in a different cohort that initially assessed.  I strongly recommend to recalculate the results including individual with CVD. Maybe it is also worth to compare AF incidence among patients with and without AF at the outset?
  2. There are too many details provided in the methods section as the study protocol has been described elsewhere [Peasey at al. Determinants of cardiovascular disease and other non-communicable diseases in Central and Eastern Europe: rationale and design of the HAPIEE study]. Therefore this should be intensively trimmed.
  3. The presentation of AF incidence, both at the baseline and during the follow-up, is not clear. I recommend to add absolute numbers to percentages i.e,
    a) prevalence of AF at the outset  was 1.6% {145/9360}, b) there were 122 new cases of AF identified,  accounting for 3.2%  {122/3871}
  1. According to 2020 ESC Guidelines for the diagnosis and management of atrial fibrillation the diagnosis of AF requires rhythm documentation with an ECG tracing showing AF. However only an episode lasting at least 30 s is diagnostic for clinical AF (class I level B). It is another limitation for the presented study as AF detection was limited to a single 12-lead ECG registration (probably 5sec registration, however timeframe has not been provided) that almost certainly detected non-clinical AF. It should be intensively discussed in the manuscript and add to the study limitations.

Author Response

Please see the attachment file with the response to the reviewer’s comments

Reviewer 2 Report

Overall quality of current manuscript is quite fine.

There are some important POINTs of STREGHT such as statistics.

On the contrary, I'd like to underline SOME MINOR REVISIONS to authors:

  • A fast check for English language. Basically good. Only minor spelling or typing errors
  • Moreover, I think the abbreviation ACVD (line 16 and following in the text) is not totally adequate. International coding is generally ASCVD. PLEASE CHECK.
  • I'd like to receive some further information (and eventually also references) about the cut-off choice for middle-aged and older people. PLEASE BETTER EXPLAIN.
  • Analogously, why a 13-years study? Atypical for duration. PLEASE CLARIFY.
  • And finally, I'd like to suggest a strongly enunciation of study limitations in CONCLUSIONS.

With my best regards.

Author Response

(The authors gave the same response as above.)

Round 2

Reviewer 1 Report

The authors have implemented all suggestions sufficiently. The key message of the paper is now clearly transported.  However it seems to me that the manuscript needs extensive revision for language and grammar.

Author Response

Please see the attachment file with author's reply to the review report 
